# Oral Health of Patients Undergoing Percutaneous Coronary Intervention—A Possible Link between Periodontal Disease and In-Stent Restenosis

**DOI:** 10.3390/jpm13050760

**Published:** 2023-04-28

**Authors:** Ferenc Tamás Nagy, Dorottya Gheorghita, Lalli Dharmarajan, Gábor Braunitzer, Alexandru Achim, Zoltán Ruzsa, Márk Ádám Antal

**Affiliations:** 1Division of Invasive Cardiology, Department of Internal Medicine, University of Szeged, 6720 Szeged, Hungary; nagy.ferenc.ikr@gmail.com (F.T.N.);; 2Faculty of Dentistry, Department of Esthetic and Operative Dentistry, University of Szeged, 6720 Szeged, Hungary; 3Private Practice, Chennai 600089, Tamil Nadu, India; 4dicomLAB Dental, Ltd., 7522 Szeged, Hungary; 5“Nicolae Stancioiu” Heart Institute, “Iuliu Hatieganu” University of Medicine and Pharmacy, 400012 Cluj-Napoca, Romania

**Keywords:** in-stent restenosis, percutaneous coronary intervention, periodontal disease

## Abstract

***Introduction:*** There is a well-documented association between coronary artery disease (CHD) and periodontal disease (PD) mediated by common inflammatory pathways. This association, however, has not been investigated extensively in the special context of in-stent restenosis. This study aimed to investigate the periodontal status of patients undergoing percutaneous coronary intervention (PCI) for restenotic lesions. ***Methods and Results***: We enrolled 90 patients undergoing percutaneous coronary intervention and 90 age- and gender-matched healthy controls in the present study. All subjects received a full-mouth examination by a periodontist. Plaque index, periodontal status, and tooth loss were determined. The periodontal state was significantly worse (*p* < 0.0001) in the PCI group, and each periodontal stage increased the odds of belonging to the PCI group. This effect of PD was independent of diabetes mellitus, another strong risk factor for CAD. The PCI group was further divided into two subgroups: PCI for restenotic lesions (*n* = 39) and PCI for de novo lesions (*n* = 51). Baseline clinical and procedural characteristics were comparable between the two PCI subgroups. A significant (*p* < 0.001) association was found between the PCI subgroup and the severity of periodontal disease, with the incidence of severe PD reaching 64.1%. ***Conclusions:*** Patients undergoing PCI for in-stent restenosis exhibit more severe forms of periodontal disease not only as compared to healthy controls but also as compared to patients stented for de novo lesions. The potential causality between PD and restenosis must be studied in larger prospective studies.

## 1. Introduction

Coronary artery disease (CAD) remains the leading cause of morbidity and mortality worldwide. Over the past 40 years, percutaneous coronary intervention (PCI) and stent implantation have revolutionized the field of coronary artery disease (CAD) treatment. Owing to its importance, interventional cardiology remains a field of intensive research and development [1]. Despite technical advances and the development of newer-generation drug-eluting stents (DES), in-stent restenosis (ISR) remains the Achilles heel of interventional cardiology. ISR rates range from 3 to 30% depending on the type of stent used and the clinical setting [2]. Risk factors for ISR are diverse and include patient, lesion, and procedural characteristics. The most important patient characteristics and systemic disease states associated with ISR are age, female gender, diabetes, and chronic kidney disease [3].

Periodontal disease (PD) is defined as a bacterially induced, chronic inflammation of the supporting tissues of the teeth [4]. It is among the most common diseases worldwide, with up to 11% prevalence in its severe form [5]. PD and cardiovascular diseases are both related to several common risk factors, and an increasing number of epidemiological studies seem to support the hypothesis of a possible association between periodontal disease and atherosclerosis [6,7,8]. The role of inflammatory processes is well documented in the pathogenesis of both atherosclerosis and periodontal disease, most likely proving to be the link between the two diseases. Infected periodontal pockets are typical reservoirs of such organisms and their toxins and degradation products, which can cause inflammatory, immunological, and humoral activities, enter the bloodstream, and contribute to atherogenesis and thromboembolic events [9]. 

There are several possible pathways in which infectious agents may affect atherosclerotic processes. In periodontitis, bacterial plaque impairs the periodontal epithelium and allows the entry of oral pathogens and their harmful elements (endotoxins and exotoxins) into the bloodstream. Direct invasion of the vessel wall by oral pathogens can trigger an inflammatory response that leads to endothelial dysfunction. This induces infiltration of inflammatory cells and vascular smooth muscle cell proliferation, which constitute the pathogenesis of atherosclerosis [10]. 

Although the exact pathophysiological pathways leading to de novo coronary artery stenosis, bare metal stent restenosis (BMS), and DES restenosis are not the same, inflammation plays an important role in both [11]. This raises the possibility that periodontal disease linked to a systemic inflammatory response may be associated with not only de novo coronary stenosis but in-stent restenosis as well. A possible association beyond clinical interest may have therapeutic implications as well. From the periodontal point of view, it is also necessary to consider the updates in the non-invasive treatment of periodontitis and in the maintenance of the periodontal patient over the years, which can lead to the maintenance of a healthy condition or to a considerable slowdown in the progression of the periodontal pathology, which consistently results in a lower concentration of inflammatory tissue mediators frequently involved in these processes [12,13].

The current study aimed to ascertain any possible connections between periodontal disease and cardiovascular status in PCI patients. We also hypothesized that periodontal disease might be more prominent in patients undergoing PCI for restenosis, thereby linking periodontal status to the development and progression of restenosis.

## 2. Materials and Methods

### 2.1. Ethics Statement

The study was approved by the Human Ethics Review Board of the University of Szeged (approval No. 127/2015-SZTE), and the study design conformed to the Declaration of Helsinki in all respects. Written informed consent was obtained from all participants. 

### 2.2. Protocol and Study Participants

The hospital-based case-control study was conducted between 2016 and 2017. Participants (*n* = 90) were selected from patients who underwent PCI at the Invasive Cardiology Unit of the Internal Medicine Department, University of Szeged. Of the 90 patients, 51 underwent coronary intervention for de novo lesions and 39 for in-stent restenosis. The healthy control group (*n* = 90) consisted of people attending mandatory lung screening in the same city and period. Participation was voluntary in both groups; the volunteers did not receive any kind of compensation and were free to quit at any time. PCI and control groups were gender- and age-matched.

The indication for percutaneous coronary intervention was set up by an interventional cardiologist before the beginning of this study in all cases according to local practice based on ESC guidelines [14]. Patients with restenosis were eligible for the study if they were diagnosed with in-stent restenosis (defined as re-narrowing of ≥50% of the vessel diameter inside or immediately adjacent to the stent) by an experienced cardiologist requiring coronary intervention [11]. Medical information of each group was extracted from patient files and hospital records. Laboratory values were acquired as a part of the general post-PCI workup. A questionnaire collected demographic and tobacco use data. Participants were grouped as smokers and non-smokers based on their self-reported tobacco use. 

Patients with non-cardiovascular diseases influencing periodontal status, such as excessive alcohol consumption, drug abuse, estrogen deficiency, and local or systemic inflammatory conditions, were excluded from the study. Critically ill patients and patients with less than four teeth were also excluded from the study. 

The clinical staging of periodontal disease is currently a subject of debate, even though the progression of the disease is well-established in pathological terms [15]. In the present study, the staging proposed by Fernandes and colleagues was used [16]. It requires the following indicators to be recorded: the number of missing teeth (excluding third molars), plaque index [PI, also known as the Silness-Löe Index (0–3)], bleeding on probing (BOP; the presence or absence of bleeding within 15 s after probing), probing pocket depth (PPD; in millimeters), and clinical attachment level (CAL; to describe the position of the soft tissue in relation to the cemento–enamel junction). This protocol has already been used by this research group in several publications [17,18,19]. According to the above, a patient’s periodontal status is reported as belonging to one of four subgroups: (1) healthy, (2) early, (3) moderate, or (4) severe periodontal disease.

### 2.3. Clinical Examination

Each subject received a full mouth periodontal examination within 48 h after the percutaneous coronary intervention, performed by an expert in the field. PPD, CAL, and BOP were examined with Williams probes (Hu-Friedy Manufacturing Co., Chicago, IL, USA) at six sites per tooth (mesiobuccal, buccal, distobuccal, disto-lingual, lingual, mesio-lingual). 

### 2.4. Statistical Analysis

Statistical analyses were performed in SPSS 21.0 (IBM, Armonk, NY, USA). Continuous variables were described as means and standard deviations; categorical variables were characterized with frequencies unless otherwise stated. For the comparison of specific parameters between the groups, either one-way ANOVA (with Tukey’s HSD for the pairwise comparisons), Kruskal–Wallis ANOVA (with Mann–Whitney U for the pairwise comparisons), or the chi-square test was used, depending on the characteristics of the dataset. The normality of the datasets was determined with the Shapiro–Wilk test, and Levene’s test was used to test for the homogeneity assumption. The general significance limit was set at *p* = 0.05 but was corrected for multiple comparisons with the Bonferroni correction where necessary. For hypothesis testing, logistic and multinomial logistic regression analyses and the chi-square test was used. 

## 3. Results

The clinical characteristics of the PCI and healthy control groups are presented in Table 1, with the results of the between-groups comparisons for the different parameters. Figure 1 shows the distribution of periodontitis severity categories in the three groups (PCI patients with restenotic lesions, PCI patients with de novo lesions, and controls). The chi-square test indicated a significant association between group and periodontal status (χ^2^ = 35,207, df = 6, *p* < 0.001). That is, whether a subject belonged to the PCI or the healthy control group was significantly associated with their periodontal status. We also found a significant difference in the number of teeth (*p* < 0.001) but not in plaque index between the two groups.

To examine this question further, a logistic regression model was built. In this model, background variables potentially interfering with systemic health in themselves and/or interfering with the health of the periodontal tissues (periodontal status, diabetes mellitus, smoking, plaque index) were included as factors, and the group (patient or control) was the dependent variable. With this analysis, we sought to determine which factors had a significant association with a subject belonging to the PCI group. The analysis indicated that diabetes mellitus (*p* < 0.001) and periodontal status (*p* < 0.05) were such factors. As for diabetes mellitus, we hypothesized that this factor could influence periodontal health (as often reported in the literature), but this could not be verified in this sample. The chi-square test indicated no significant association (χ^2^ = 3.049, df = 3, *p* = 0.384), so we concluded that in this sample, periodontal health was not influenced by the presence or absence of diabetes. In other words, the two factors exerted their significant effect independently. Regarding periodontal status, we also sought to determine if belonging to the patient group was significantly associated with the severity (stage) of periodontal disease. For this, we used multinomial regression analysis with the group as the target variable and periodontal status as the independent variable. The multinomial regression analysis confirmed the significant effect of periodontal status on whether a given subject belonged to the patient or control group (*p* < 0.001). The odds ratios were as follows: ORstage2: 2.154 (95%CI: 0.42–11.08), ORstage3: 3.13 (95% CI: 0.61–16.04), ORstage4: 15.27 (95%CI: 2.83–82.42). Each periodontal stage increased the odds of belonging to the patient group, but the difference was significant only in the most severe stage (stage 4, *p* < 0.01). In other words, if one had stage 4 periodontal disease, that person had a significantly higher chance of being found in the group of cardiovascular patients at an odds ratio of 15.27. 

Within the PCI patient group, two subgroups were differentiated according to the lesion treated during PCI. Fifty-one patients underwent PCI for de novo lesions and 39 for in-stent restenotic lesions. Comparative descriptive data of these groups are given in Table 2, with the results of the between-groups comparisons for the different parameters. We found no significant differences between the two groups regarding baseline clinical and procedural characteristics. Although the sample size was not sufficient for any deeper comparison of these groups, we still wanted to know if the severity of periodontal disease might be associated with restenosis. The frequencies of the four stages of PD by patient subgroup are presented in Figure 1. The χ^2^ test indicated a significant association between the patient subgroup and the severity of PD (χ^2^ = 13.77, df = 3, *p* < 0.01). Figure 1 shows that the incidence of the individual stages is rather disproportionate in the restenosis group: 64.1% of all the cases fall in stage 4. In the de novo stenosis group, the cases were much more evenly distributed across the categories, even if no healthy case was observed. There was no significant difference between the two groups regarding the number of teeth or plaque index.

## 4. Discussion

The present study investigated the periodontal status of patients undergoing percutaneous coronary intervention. Patients undergoing PCI for significant coronary artery lesions were more likely to have moderate or severe periodontitis than healthy controls. We also found significant differences in the periodontal health indices between patients undergoing PCI for restenosis and de novo lesions. Namely, severe forms of PD were more often found among patients with restenosis. Our results raise the possibility of an association between in-stent restenosis and periodontal disease. 

### 4.1. Periodontal Disease and CAD

The link between periodontal disease and angiographically verified coronary disease has been well established in stable coronary artery disease and acute coronary syndromes [20,21,22,23]. Nonetheless, it is still a matter of debate whether PD is an independent risk factor for CAD or whether the association is based on an abundance of shared risk factors, including, amongst others: age, gender, smoking, diabetes mellitus, hypertension, obesity, low socioeconomic status, and stress [24]. In our age- and gender-matched case-control study, we were not only able to provide further evidence for a strong association between PD and CAD, but we also showed that this association is independent of diabetes mellitus (one of the most prominent shared risk factors for both CAD and PD).

While there was a clear difference in PD status, no meaningful difference was found in plaque index between patients undergoing PCI and healthy controls. The relationship between oral hygiene, as represented by PI, and periodontal disease is well established [25]. Furthermore, hospitalization itself (for the intervention) might negatively affect oral self-care and PI [26,27]. In this study, we found no such effect, so it is unlikely that the difference in PD status was mediated by poor oral hygiene. 

### 4.2. Periodontal Disease and Restenosis

Periodontal disease in the special subset of CAD patients with restenosis has not been studied extensively. Fukushima et al. found an association between PD at baseline and increased risk of future major adverse cardiac events (MACE) in CAD patients who underwent PCI with a drug-eluting stent (DES) for de novo coronary lesions [28]. In this study, however, MACE was driven by non-target lesion myocardial infarction and death and was not powered to investigate differences in restenosis occurrence. Wu Y et al. found a positive, independent correlation between triglyceride index (a marker of insulin resistance), oral infections, and in-stent restenosis in the subgroup of acute coronary syndrome patients 24 months after DES implantation [29]. 

The pathophysiology of restenosis differs greatly from that of de novo coronary artery disease [11]. Restenosis is a local response to vascular injury caused by the coronary intervention itself in the form of balloon dilatation and/or stent implantation. Damage to the arterial wall triggers an inflammatory reaction which plays a key role in the activation, migration, and proliferation of endothelial, smooth muscle cells, and macrophages [30]. This, in turn, leads not only to reendothelization but potentially neointimal hyperplasia and, in the long run, neoatherosclerosis, thus re-narrowing the stented lumen [31]. Our findings raise the novel possibility of chronic periodontal disease playing a role in this specific pathologic subset, leading to restenosis progression. 

Although the exact mechanisms are unknown, several pathways have been suggested by which periodontal disease may contribute to the progression of in-stent restenosis [29]. The most plausible explanation would be through the common inflammatory pathway already well recognized in connecting CAD progression and PD [9]. Indeed, markers of inflammation that are elevated in PD, such as CRP, matrix metalloproteinase 2, and TNF-α, can also increase the risk of restenosis [32]. PD is also closely associated with endothelial dysfunction, a known risk factor of ISR [33]. Endothelial dysfunction in PD may occur as a result of direct vessel wall invasion by oral pathogens triggering an inflammatory response [34] or possibly through high levels of circulating trimethylamine N-oxide, a harmful oral microbiota-generated metabolite [35]. Platelets play an important role not only in hemostasis but also in inflammation. Periodontal pathogens, such as Porphyromonas gingivalis, activate platelets and cause platelet aggregation through HgP44 (hemagglutinin domain protein) [36]. Activated platelets, in turn, can contribute not only to atherosclerosis but also to the progression of restenosis [37]. Furthermore, Porphyromonas gingivalis may influence gene expression in vascular smooth cells leading to increased proliferation and may also induce migration [38]. Smooth muscle cell proliferation and migration play an important role in the pathogenesis of neointimal hyperplasia leading to ISR [30]. Finally, it is important to note that as with PD and atherosclerosis, PD and in-stent restenosis also have been associated with common underlying medical conditions, such as age, gender, diabetes mellitus, chronic kidney disease, and multivessel coronary disease. [3,24] Thus, the question of causality, whether PD is an independent risk factor of in-stent restenosis or the association is based on common risk factors, is a possible debate as with PD and atherosclerosis [24]. 

### 4.3. Clinical Implications

Current guidelines advise the use of local antiproliferative agents via drug-eluting stents and balloons in the prevention and treatment of in-stent restenosis. They also stress the importance of intravascular imaging to recognize possible underlying mechanical substrates and advise interventional treatment algorithms based on these results [39]. However, besides local drug- and mechanical therapy, considerable research is directed toward preventing restenosis via a systemic approach using anti-inflammatory drugs. Evidence on the effect of periodontal therapy on surrogate risk factors of systemic inflammation (IL6, CRP, TNF-alpha) is well established [23,40,41]. This raises the intriguing possibility of systemic inflammation inhibition by periodontal therapy playing a role in a multidisciplinary approach to the prevention of in-stent restenosis. Further prospective randomized clinical trials would be needed to prove this possible effect. On a more general level, our findings call attention to the importance of screening for poor oral hygiene and already-developed periodontal disease in the high cardiovascular-risk group of patients undergoing coronary interventions, especially those with restenosis. 

### 4.4. Study Limitations 

All limitations inherent by the nature of cross-sectional, case-control studies apply to our study as well, and as such, it cannot confirm causality. Patient enrollment was skewed towards including patients undergoing PCI for restenosis. Thus, investigator bias leading to unrecognized confounding factors potentially influencing results cannot be ruled out. Patients undergoing PCI for restenosis were compared to healthy individuals and other PCI patients with comparable clinical and procedural characteristics. Therefore, lesion-specific and procedural aspects of the initial coronary intervention, which may also influence the development of restenosis, could not be considered. There were no significant differences in baseline clinical characteristics between the in-stent restenosis and de novo lesion PCI groups. This, however, does not rule out the effect of possible common underlying medical conditions as a formal multivariate analysis could not be performed due to the small number of patients. 

## 5. Conclusions

In conclusion, our study shows that patients undergoing PCI for restenotic lesions exhibit more severe forms of periodontal disease not only as compared to healthy controls but also as compared to patients stented for de novo lesions. The potential causality between PD and restenosis needs to be studied in larger prospective studies. Nonetheless, these facts further support the importance of periodontal screening and periodontal care in PCI patients, which may play a role in hindering future cardiovascular events, including restenosis.

## Figures and Tables

**Figure 1 jpm-13-00760-f001:**
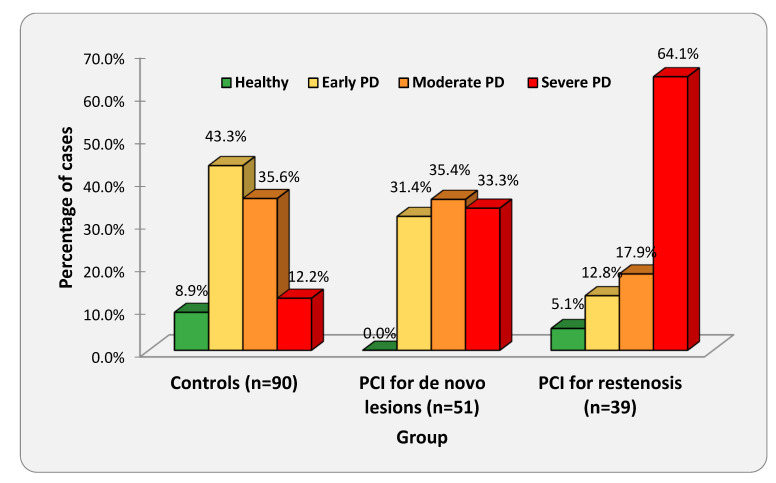
Periodontal state of patients undergoing PCI for restenotic lesions versus de novo lesions and controls.

**Table 1 jpm-13-00760-t001:** Clinical and periodontal characteristics of patients undergoing PCI versus controls.

Parameter	PCI Patients *n* = 90	Controls *n* = 90	Sig. (*p*)
Age	63.2 ± 10.0	67.3 ± 9.9	0.89
Gender (male)	63 (70)	63 (70)	1
Diabetes mellitus	29 (32)	4 (4)	<0.0001
Hypertension	82 (92)	12 (13)	<0.0001
Hyperlipidemia	67 (72)	0 (0)	<0.0001
Chronic renal disease	10 (11)	0 (0)	<0.0001
Active smoker	16 (18)	24 (27)	0.21
Former smoker	18 (20)	13 (14)	0.43
Number of teeth	14.08 ± 6.99	19.71 ± 5.73	<0.0001
Plaque index (0–3)	2 (1–3)	2 (1–2)	0.40
Periodontal status (1–4)	3 (2–4)	2 (2–3)	<0.0001

Significance limit: *p* = 0.05. Numerical values are presented as average ± standard deviation. Categorical values are presented as *n* (%). Plaque index and periodontal status as median (25–75th quartile); PCI—percutaneous coronary intervention.

**Table 2 jpm-13-00760-t002:** Clinical and periodontal characteristics of patients undergoing PCI for restenotic lesions versus de novo lesions Significance limit: *p* = 0.05. Numerical values are presented as average ± standard deviation or median (25–75th quartile) as appropriate. Categorical values are presented as *n* (%). ACE—angiotensin-converting enzyme; ACS—acute coronary syndrome; ARB—angiotensin II receptor blocker; BMI—body mass index; BUN—blood urea nitrogen; CKD—chronic kidney disease; LAD—left anterior descending artery; EF— ejection fraction; LCX—left circumflex artery; LMT—left main trunk; LV—left ventricular; MI—myocardial infarction; MPV—mean platelet volume; (N)OAC—(new) oral anticoagulant; PCI—percutaneous coronary intervention; RCA—right coronary artery; WBC—white blood cell; *in 7 cases DEB was used during PCI for restenosis.

Parameter	PCI Patients with Restenotic Lesions (*n* = 39)	PCI Patients with De Novo Lesions (*n* = 51)	Sig. (*p*)
**Demography**			
Age (years)	63.7 ± 8.9	62.8 ± 10.9	0.67
Gender (male)	27 (69)	36 (70.6)	1.00
BMI	30 ± 5.5	29.4 ± 4.7	0.77
Active smoker	6 (15)	10 (20)	0.06
Former smoker	9 (23)	9 (18)	0.43
**Medical history**			
Hypertension	35 (95)	47 (92)	0.83
Diabetes mellitus	14 (36)	15 (29)	0.64
CKD	5 (13)	5 (10)	0.74
Hyperlipidemia	32 (82)	35 (69)	0.14
Prior PCI	39 (100)	27 (52.9)	NA
LV-EF	55.4 ± 14.5	56.2 ± 10.9	0.99
**Medications**			
Aspirin	38 (97)	50 (98)	0.84
P2Y12 inhibitor	39 (100)	51 (100)	1
(N)OAC	9 (23)	8 (16)	0.72
Beta-blocker	34 (87)	47 (92)	0.44
ACE inhibitor/ARB	35 (90)	47 (92)	0.69
Statin	36 (92)	48 (94)	0.73
Ca-channel blocker	7 (18)	16 (31)	0.15
PPI	35(90)	43 (84)	0.45
**Laboratory data**			
WBC (×10^3^/μL)	7.6 ± 1.9	7.9 ± 1.7	0.2
Hemoglobin (g/dL)	125.3 ± 27	131.8 ± 12.9	0.65
Thrombocyte (×10^3^/μL)	208.6 ± 67.2	219.3 ± 52.1	0.18
MPV (fL)	10.9 ± 0.7	10.7 ± 1	0.33
Creatinine (μmol)	91 ± 38.9	87.9± 27.1	0.78
**Procedural data**			
LM/LAD	19 (40)	26 (42)	0.84
LCX	9 (18)	19 (31)	0.13
RCA	20 (42)	17 (27)	0.22
Multivessel PCI	16 (41)	20 (39)	0.86
Stent length	46.6 ± 29.2	40.2 ± 26.9	0.42
Stent diameter	3.1 ± 0.4	2.98 ± 0.47	0.16
DES*	32 (82)	51 (100)	NA
**Periodontal condition**			
Number of teeth	13.8 ± 7	14.3 ± 7.1	0.74
Plaque index (0–3)	2 (1–3)	2 (1–3)	0.83
Periodontal status (1–4)	4 (3–4)	3 (2–4)	<0.01

## Data Availability

The dataset is available from the corresponding author upon reasonable request.

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
