# Peer review of "Oral Health of Patients Undergoing Percutaneous Coronary Intervention—A Possible Link between Periodontal Disease and In-Stent Restenosis"

_jpm, 2023, doi:10.3390/jpm13050760_

Round 1

Reviewer 1 Report

The authors could discuss better the presence of interaction with health conditions ins this population, because it was not tested in a multivariate way, only in a bivariate analysis. Could discuss better too the limits of the study population

Author Response

Thank you for your valuable feedback. We appreciate your suggestion to discuss the presence of interaction with health conditions in this population in more detail. As you have rightly pointed out, we did not test this interaction in a multivariate analysis, and we agree that this is an important consideration. However, the various health conditions were so unevenly distributed in the sample that it was the reason we did not follow that path. Although technically possible, an analysis of this kind would be questionable given the sample size, and the conclusions based on such analyses would border on speculation. Therefore, we chose to provide only a descriptive analysis of these conditions to allow readers to interpret the results with the possible effect of these conditions in mind.

We acknowledge that this is a limitation of our study, and we have updated our manuscript to reflect this limitation and provide a more nuanced discussion of the study population.

Reviewer 2 Report

Dear Authors, 

you made a great work! However, some improvements are suggested to improve the manuscript quality.

The paper is an original research study on the Oral health status of patients undergoing percutaneous coronary intervention.

The Authors made a great work in terms of methodology and the paper sounds scientific and well written.

However, some improvements are mandatory before acceptance.

The abstract is well written, complete and summary in its various aspects. The keywords are complete and appropriate.

In the introduction:

·       “Periodontal disease (PD) is defined as a bacterially induced, chronic inflammation of the supporting tissues of the teeth. [4] It is among the most common diseases worldwide with up to 11% prevalence in its severe form. [5]” From the periodontal point of view, it is also necessary to consider the updates in the non-invasive treatment of periodontitis, and in the maintenance of the periodontal patient over the years, which can lead to the maintenance of a healthy condition or to a considerable slowdown in the progression of the periodontal pathology and peri -implant, which consistently results in a lower concentration of tissue inflammatory mediators frequently involved in these processes, as indicated by:

" Balaji TM, Varadarajan S, Jagannathan R, Mahendra J, Fageeh HI, Fageeh HN, Mushtaq S, Baeshen HA, Bhandi S, Gupta AA, Raj AT, Reda R, Patil S, Testarelli L. Melatonin as a Topical/Systemic Formulation for the Management of Periodontitis: A Systematic Review. Materials (Basel). 2021 May 6;14(9):2417. doi: 10.3390/ma14092417."
“Guarnieri R, Zanza A, D'Angelo M, Di Nardo D, Del Giudice A, Mazzoni A, Reda R, Testarelli L. Correlation between Peri-Implant Marginal Bone Loss Progression and Peri-Implant Sulcular Fluid Levels of Metalloproteinase-8. J Pers Med. 2022 Jan 6;12(1):58. doi: 10.3390/jpm12010058.”

Materials and methods are clear and well explained. Different aspects are analyzed with a dedicated statistical test. The authors did a great job in the explication of all the variables identified and included in the study.

Results are easy to understand and comprehensive. The tables are clearly legible and complete. All the studied characteristics were reported in tables which are, moreover, clear and concise. The text is clear and well articulated.

In the discussion:

·       “Indeed, markers of inflammation that are elevated in PD such as CRP, matrix metalloproteinase 2, and TNF-α can also increase the risk of restenosis. [30]” this assessment reflects the one I previously placed in the introduction, and I congratulate the authors for having had the sensitivity and knowledge to include this interesting topic.

·       this section is complete and evaluates the outcome of different papers present in literature. The overall is comprehensive, concise and complete in its various aspects.

Conclusions are concise and clear.

Bibliography should be formatted respecting the journal’s requirements and no improper citations are evidenced.

English is clear and easy to understand.

Author Response

Thank you for pointing out the importance of advances in periodontal therapy and periodontal therapy itself in maintenance and slowing down progression of PD which results in lower concentration of inflammatory mediators. Your suggestions were incorporated in the manuscript with the appropriate citations.

Please see below:

„A possible association beyond clinical interest may have therapeutic implications as well. As from the periodontal point of view, it is also necessary to consider the updates in the non-invasive treatment of periodontitis, and in the maintenance of the periodontal patient over the years, which can lead to the maintenance of a healthy condition or to a con-siderable slowdown in the progression of the periodontal pathology, which consistently results in a lower concentration of tissue inflammatory mediators frequently involved in these processes. [12-13]”

 Also please note, bibliography has been formatted respecting the journal’s requirements without any improper citations and the sugested citations have been also added.

Round 2

Reviewer 2 Report

Dear Authors, 

Thank you for made the suggested improvements. 

You made a great work!